# FEDERATED USER REPRESENTATION LEARNING

## ABSTRACT

Collaborative personalization, such as through learned user representations (embeddings), can improve the prediction accuracy of neural-network-based models significantly. We propose *Federated User Representation Learning* (FURL), a simple, scalable, privacy-preserving and resource-efficient way to utilize existing neural personalization techniques in the Federated Learning (FL) setting. FURL divides model parameters into *federated* and *private* parameters. *Private* parameters, such as private user embeddings, are trained locally, but unlike *federated* parameters, they are not transferred to or averaged on the server. We show theoretically that this parameter split does not affect training for most model personalization approaches. Storing user embeddings locally not only preserves user privacy, but also improves memory locality of personalization compared to on-server training. We evaluate FURL on two datasets, demonstrating a significant improvement in model quality with 8% and 51% performance increases, and approximately the same level of performance as centralized training with only 0% and 4% reductions. Furthermore, we show that user embeddings learned in FL and the centralized setting have a very similar structure, indicating that FURL can learn collaboratively through the shared parameters while preserving user privacy.

## 1 INTRODUCTION

Collaborative personalization, like learning user embeddings jointly with the task, is a powerful way to improve accuracy of neural-network-based models by adapting the model to each user's behavior (Grbovic & Cheng, 2018; Ni et al., 2018; Lee et al., 2017; Jaech & Ostendorf, 2018; McGraw et al., 2016; Vosecky et al., 2014). However, model personalization usually assumes the availability of user data on a centralized server. To protect user privacy, it is desirable to train personalized models in a privacy-preserving way, for example, using Federated Learning (McMahan et al., 2016; Konen et al., 2016b). Personalization in FL poses many challenges due to its distributed nature, high communication costs, and privacy constraints (Li et al., 2019a; Bonawitz et al., 2019; Caldas et al., 2018; Li et al., 2019b; 2018; Liu et al., 2018; Yang et al., 2019; Konen et al., 2016a).

To overcome these difficulties, we propose a simple, communication-efficient, scalable, privacy-preserving scheme, called FURL, to extend existing neural-network personalization to FL. FURL can personalize models in FL by learning task-specific user representations (i.e., embeddings) (Lerer et al., 2019; Grbovic & Cheng, 2018; Ni et al., 2018; Lee et al., 2017; Jaech & Ostendorf, 2018) or by personalizing model weights (Tang & Wang, 2018). Research on collaborative personalization in FL (Smith et al., 2017; Sebastian Caldas, 2019; Chen et al., 2018; Yao et al., 2019) has generally focused on the development of new techniques tailored to the FL setting. We show that most existing neural-network personalization techniques, which satisfy the *split-personalization* constraint (1,2,3), can be used directly in FL, with only a small change to Federated Averaging (McMahan et al., 2016), the most common FL training algorithm.

Existing techniques do not efficiently train user embeddings in FL since the standard Federated Averaging algorithm (McMahan et al., 2016) transfers and averages all parameters on a central server. Conventional training assumes that all user embeddings are part of the same model. Transferring all user embeddings to devices during FL training is prohibitively resource-expensive (in terms of communication and storage on user devices) and does not preserve user privacy.

FURL defines the concepts of *federated* and *private* parameters: the latter remain on the user device instead of being transferred to the server. Specifically, we use a *private* user embedding vector on

each device and train it jointly with the global model. These embeddings are never transferred back to the server.

We show theoretically and empirically that splitting model parameters as in FURL affects neither model performance nor the inherent structure in learned user embeddings. While global model aggregation time in FURL increases linearly in the number of users, this is a significant reduction compared with other approaches (Smith et al., 2017; Sebastian Caldas, 2019) whose global aggregation time increases quadratically in the number of users.

FURL has advantages over conventional on-server training since it exploits the fact that models are already distributed across users. There is little resource overhead in distributing the embedding table across users as well. Using a distributed embeddings table improves the memory locality of both training embeddings and using them for inference, compared to on-server training with a centralized and potentially very large user embedding table.

Our evaluation of document classification tasks on two real-world datasets shows that FURL has similar performance to the server-only approach while preserving user privacy. Learning user embeddings improves the performance significantly in both server training and FL. Moreover, user representations learned in FL have a similar structure to those learned in a central server, indicating that embeddings are learned independently yet collaboratively in FL.

In this paper, we make the following contributions:

- We propose FURL, a simple, scalable, resource-efficient, and privacy preserving method that enables existing collaborative personalization techniques to work in the FL setting with only minimal changes by splitting the model into *federated* and *private* parameters.

- We provide formal constraints under which the parameter splitting does not affect model performance. Most model personalization approaches satisfy these constraints when trained using Federated Averaging (McMahan et al., 2016), the most popular FL algorithm.

- We show empirically that FURL significantly improves the performance of models in the FL setting. The improvements are 8% and 51% on two real-world datasets. We also show that performance in the FL setting closely matches the centralized training with small reductions of only 0% and 4% on the datasets.

- Finally, we analyze user embeddings learned in FL and compare with the user representations learned in centralized training, showing that both user representations have similar structures.

## 2 RELATED WORK

Most existing work on collaborative personalization in the FL setting has focused on FL-specific implementations of personalization. Multi-task formulations of Federated Learning (MTL-FL) (Smith et al., 2017; Sebastian Caldas, 2019) present a general way to leverage the relationship among users to learn personalized weights in FL. However, this approach is not scalable since the number of parameters increases quadratically with the number of users. We leverage existing, successful techniques for on-server personalization of neural networks that are more scalable but less general, i.e., they satisfy the *split-personalization* constraint (1,2,3).

Transfer learning has also been proposed for personalization in FL (Hartmann, 2018), but it requires alternative freezing of local and global models, thus complicating the FL training process. Moreover, some versions (Zhao et al., 2018) need access to global proxy data. Chen et al. (2018) uses a two-level meta-training procedure with a separate query set to personalize models in FL.

FURL is a scalable approach to collaborative personalization that does not require complex multi-phase training, works empirically on non-convex objectives, and leverages existing techniques used to personalize neural networks in the centralized setting. We show empirically that user representations learned by FURL are similar to the centralized setting. Collaborative filtering (Ammad-ud-din et al., 2019) can be seen as a specific instance of the generalized approach in FURL. Finally, while fine-tuning individual user models after FL training (Popov & Kudinov, 2018) can be effective, we focuses on more powerful collaborative personalization that leverages common behavior among users.

## 3 LEARNING PRIVATE USER REPRESENTATIONS

The main constraint in preserving privacy while learning user embeddings is that embeddings should not be transferred back to the server nor distributed to other users. While typical model parameters are trained on data from all users, user embeddings are very privacy-sensitive (Resheff et al., 2018) because a user's embedding is trained only on that user's data.

FURL proposes splitting model parameters into *federated* and *private* parts. In this section, we show that this parameter-splitting has no effect on the FL training algorithm, as long as the FL training algorithm satisfies the *split-personalization* constraint. Models using common personalization techniques like collaborative filtering, personalization via embeddings or user-specific weights satisfy the *split-personalization* constraint when trained using Federated Averaging.

### 3.1 SPLIT PERSONALIZATION CONSTRAINT

FL algorithms typically have two steps:

1. *Local Training*: Each user initializes their local model parameters to be the same as the latest global parameters stored on the server. Local model parameters are then updated by individual users by training on their own data. This produces different models for each user.[1]
2. *Global Aggregation*: Locally-trained models are "aggregated" together to produce an improved global model on the server. Many different aggregation schemes have been proposed, from a simple weighted average of parameters (McMahan et al., 2016), to a quadratic optimization (Smith et al., 2017).

To protect user privacy and reduce network communication, user embeddings are treated as private parameters and not sent to the server in the aggregation step. Formal conditions under which this splitting does not affect the model quality are described as follows.

Suppose we train a model on data from $n$ users, and the $k$-th training example for user $i$ has features $x_k^i$, and label $y_k^i$. The predicted label is $\hat{y}_k^i = f(x_k^i; w_f, w_p^1, \ldots, w_p^n)$, where the model has *federated* parameters $w_f \in \mathbb{R}^f$ and *private* parameters $w_p^i \in \mathbb{R}^p \ \forall i \in \{1, \ldots, n\}$.

In order to guarantee no model quality loss from splitting of parameters, FURL requires the *split-personalization* constraint to be satisfied, i.e., any iteration of training produces the same results irrespective of whether private parameters are kept locally, or shared with the server in the aggregation step. The two following constraints are sufficient (but not necessary) to satisfy the *split-personalization* constraint: local training must satisfy the *independent-local-training* constraint (1), and global aggregation must satisfy the *independent-aggregation* constraint (2,3).

### 3.2 INDEPENDENT LOCAL TRAINING CONSTRAINT

The *independent-local-training* constraint requires that the loss function used in local training on user $i$ is independent of *private* parameters for other users $w_p^j, \forall j \neq i$. A corollary of this constraint is that for training example $k$ on user $i$, the gradient of the local loss function with respect to other users' private parameters is zero:

$$\frac{\partial L(\hat{y}_k^i, y_k^i)}{\partial w_p^j} = \frac{\partial L(f(x_k^i; w_f, w_p^1, \ldots, w_p^n), y_k^i)}{\partial w_p^j} = 0 \ , \forall j \neq i \tag{1}$$

Equation 1 is satisfied by most implementations of personalization techniques like collaborative filtering, personalization via user embeddings or user-specific model weights, and MTL-FL (Smith et al., 2017; Sebastian Caldas, 2019). Note that (1) is not satisfied if the loss function includes a norm of the global user representation matrix for regularization. In the FL setting, global regularization of the user representation matrix is impractical from a bandwidth and privacy perspective. Even in centralized training, regularization of the global representation matrix slows down training a lot, and hence is rarely used in practice (Ni et al., 2018). Dropout regularization does not violate (1). Neither does regularization of the norm of each user representation separately.

---

[1]Although not required, for expositional clarity we assume local training uses gradient descent, or a stochastic variant of gradient descent.

### 3.3 INDEPENDENT AGGREGATION CONSTRAINT

The *independent-aggregation* constraint requires, informally, that the global update step for federated parameters is independent of private parameters. In particular, the global update for federated parameters depends only on locally trained values of federated parameters, and optionally, on some summary statistics of training data.

Furthermore, the global update step for private parameters for user $i$ is required to be independent of private parameters of other users, and independent of the federated parameters. The global update for private parameters for user $i$ depends only on locally trained values of private parameters for user $i$, and optionally, on some summary statistics.

The *independent-aggregation* constraint implies that the aggregation step has no interaction terms between private parameters of different users. Since interaction terms increase quadratically in the number of users, scalable FL approaches, like Federated Averaging and its derivatives (McMahan et al., 2016; Leroy et al., 2018) satisfy the *independent-aggregation* assumption. However, MTL-FL formulations (Smith et al., 2017; Sebastian Caldas, 2019) do not.

More formally, at the beginning of training iteration $t$, let $w_f^t \in \mathbb{R}^f$ denote federated parameters and $w_p^{i,t} \in \mathbb{R}^p$ denote private parameters for user $i$. These are produced by the global aggregation step at the end of the training iteration $t-1$.

**Local Training**    At the start of local training iteration $t$, model of user $i$ initializes its local federated parameters as $u_{f,i}^t := w_f^t$, and its local private parameters as $u_{p,i}^{i,t} := w_p^{i,t}$, where $u$ represents a local parameter that will change during local training. $u_{p,i}^{i,t}$ denotes private parameters of user $i$ stored locally on user $i$'s device. Local training typically involves running a few iterations of gradient descent on the model of user $i$, which updates its local parameters $u_{f,i}^t$ and $u_{p,i}^{i,t}$.

**Global Aggregation**    At the end of local training, these locally updated parameters $u$ are sent to the server for global aggregation. Equation 2 for federated parameters and Equation 3 for private parameters must hold to satisfy the *independent-aggregation* constraint. In particular, the global update rule for federated parameters $w_f \in \mathbb{R}^f$ must be of the form:

$$w_f^{t+1} := a_f(w_f^t, u_{f,1}^t, \ldots, u_{f,n}^t, s_1, \ldots, s_n) \tag{2}$$

where $u_{f,i}^t$ is the local update of $w_f$ from user $i$ in iteration $t$, $s_i \in \mathbb{R}^s$ is summary information about training data of user $i$ (e.g., number of training examples), and $a_f$ is a function from $\mathbb{R}^{f+nf+ns} \mapsto \mathbb{R}^f$.

Also, the global update rule for private parameters of user $i$, $w_p^i \in \mathbb{R}^p$, must be of the form:

$$w_p^{i,t+1} := a_p(w_p^{i,t}, u_{p,i}^{i,t}, s_1, \ldots, s_n) \tag{3}$$

where $u_{p,i}^{i,t}$ is the local update of $w_p^i$ from user $i$ in iteration $t$, $s_i \in \mathbb{R}^s$ is summary information about training data of user $i$, and $a_p$ is a function from $\mathbb{R}^{2p+ns} \mapsto \mathbb{R}^p$.

In our empirical evaluation of FURL, we use Federated Averaging as the function $a_f$, while the function $a_p$ is the identity function $w_p^{i,t+1} := u_{p,i}^{i,t}$ (more details in Section 3.4). However, FURL's approach of splitting parameters is valid for any FL algorithm that satisfies (2) and (3).

### 3.4 FURL WITH FEDERATED AVERAGING

FURL works for all FL algorithms that satisfy the *split-personalization* constraint. Our empirical evaluation of FURL uses Federated Averaging (McMahan et al., 2016), the most popular FL algorithm.

The global update rule of vanilla Federated Averaging satisfies the *independent-aggregation* constraint since the global update of parameter $w$ after iteration $t$ is:

$$w^{t+1} = \frac{\sum_{i=1}^n u_i^t c_i}{\sum_{i=1}^n c_i} \tag{4}$$

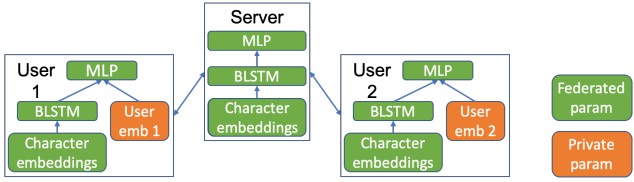

Figure 1: Personalized Document Model in FL.

where $c_i$ is the number of training examples for user $i$, and $u_i^t$ is the value of parameter $w$ after local training on user $i$ in iteration $t$. Recall that $u_i^t$ is initialized to $w^t$ at the beginning of local training. Our implementation uses a small tweak to the global update rule for private parameters to simplify implementation, as described below.

In practical implementations of Federated Averaging (Bonawitz et al., 2019), instead of sending trained model parameters to the server, user devices send model deltas, i.e., the difference between the original model downloaded from the server and the locally-trained model: $d_i^t := u_i^t - w^t$, or $u_i^t = d_i^t + w^t$. Thus, the global update for Federated Averaging in Equation 4 can be written as:

$$w^{t+1} = \frac{\sum_{i=1}^n (d_i^t + w^t) c_i}{\sum_{i=1}^n c_i} = \frac{\sum_{i=1}^n d_i^t c_i + \sum_{i=1}^n w^t c_i}{\sum_{i=1}^n c_i} = w^t + \frac{\sum_{i=1}^n d_i^t c_i}{\sum_{i=1}^n c_i} \tag{5}$$

Since most personalization techniques follow Equation 1, the private parameters of user $i$, $w_p^i$ don't change during local training on other users. Let $d_{p,i}^{j,t}$ be the model delta of private parameters of user $i$ after local training on user $j$ in iteration $t$, then $d_{p,i}^{j,t} = 0, \forall j \neq i$. Equation 5 for updating private parameters of user $i$ can hence be written as:

$$w_p^{i,t+1} = w_p^{i,t} + \frac{\sum_{j=1}^n d_{p,i}^{j,t} c_j}{\sum_{i=1}^n c_i} = w_p^{i,t} + \frac{d_{p,i}^{i,t} c_i}{\sum_{i=1}^n c_i} = w_p^{i,t} + z_i d_{p,i}^{i,t} \tag{6}$$

where $z_i = \frac{c_i}{\sum_{i=1}^n c_i}$.

The second term in the equation above is multiplied by a noisy scaling factor $z_i$, an artifact of per-user example weighting in Federated Averaging. While it is not an essential part of FURL, our implementation ignores this scaling factor $z_i$ for private parameters. Sparse-gradient approaches for learning representations in centralized training (Abadi et al., 2015; Paszke et al., 2017) also ignore a similar scaling factor for efficiency reasons. Thus, for the private parameters of user $i$, we simply retain the value after local training on user $i$ (i.e., $z_i = 1$) since it simplifies implementation and does not affect the model performance:

$$w_p^{i,t+1} = w_p^{i,t} + d_{p,i}^{i,t} = u_p^{i,t} \tag{7}$$

where $u_p^{i,t}$ is the local update of $w_p^i$ from user $i$ in iteration $t$. In other words, the global update rule for private parameters of user $i$ is to simply keep the locally trained value from user $i$.

### 3.5 FURL TRAINING PROCESS

While this paper focuses on learning user embeddings, our approach is applicable to any personalization technique that satisfies the *split-personalization* constraint. The training process is as follows:

1. Local Training: Initially, each user downloads the latest federated parameters from the server. Private parameters of user $i$, $w_p^i$ are initialized to the output of local training from the last time user $i$ participated in training, or to a default value if this was the first time user $i$ was trained. Federated and private parameters are then jointly trained on the task in question.

2. Global Aggregation: Federated parameters trained in the step above are transferred back to, and get averaged on the central server as in vanilla Federated Averaging. Private parameters (e.g., user embeddings) trained above are stored locally on the user device without being transferred back to the server. These will be used for the next round of training. They may also be used for local inference.

Figure 1 shows an example of federated and private parameters, explained further in Section 4.1

| Dataset | # Samples (Train/Eval/Test) | # Users (Train/Eval/Test) |
|---------|------------------------------|----------------------------|
| Sticker | 940K (750K/94K/96K) | 3.4K (3.3K/3.0K/3.4K) |
| Subreddit | 942K (752K/94K/96K) | 3.8K (3.8K/3.8K/3.8K) |

Table 1: Dataset statistics.

## 4 EVALUATION

We evaluate the performance of FURL on two document classification tasks that reflect real-world data distribution across users.

### 4.1 EXPERIMENTAL SETUP

**Datasets** We use two datasets, called *Sticker* and *Subreddit*. Their characteristics are as follows.

1. In-house production dataset (***Sticker***): This proprietary dataset from a popular messaging app has randomly selected, anonymized messages for which the app suggested a Sticker as a reply. The features are messages; the task is to predict user action (*click* or *not click*) on the Sticker suggestion, i.e., binary classification. The messages were automatically collected, de-identified, and annotated; they were not read or labeled by human annotators.

2. Reddit comment dataset (***Subreddit***): These are user comments on the top 256 subreddits on *reddit.com*. Following Bagdasaryan et al. (2018), we filter out users who have fewer than 150 or more than 500 comments, so that each user has sufficient data. The features are comments; the task is to predict the subreddit where the comment was posted, i.e., multiclass classification. The authors are not affiliated with this publicly available dataset (Baumgartner, 2019).

Sticker dataset has 940K samples and 3.4K users (274 messages/user on average) while Subreddit has 942K samples and 3.8K users (248 comments/user on average). Each user's data is split (0.8/0.1/0.1) to form train/eval/test sets. Table 1 presents the summary statistics of the datasets.

**Document Model** We formulate the problems as document classification tasks and use the a LSTM-based (Hochreiter & Schmidhuber, 1997) neural network architecture in Figure 1. The text is encoded into an input representation vector by using character-level embeddings and a Bidirectional LSTM (BLSTM) layer. A trainable embedding layer translates each user ID into a user embedding vector. Finally, an Multi-Layer Perceptron (MLP) produces the prediction from the concatenation of the input representation and the user embedding. All the parameters in the character embedding, BLSTM and MLP layers are *federated* parameters that are shared across all users. These parameters are locally trained and sent back to the server and averaged as in standard Federated Averaging. User embedding is considered a *private* parameter. It is jointly trained with federated parameters, but kept privately on the device. Even though user embeddings are trained independently on each device, they evolve *collaboratively* through the globally shared model, i.e., embeddings are multiplied by the same shared model weights.

**Configurations** We ran 4 configurations to evaluate the performance of the models with/without FL and personalization: *Global Server*, *Personalized Server*, *Global FL*, and *Personalized FL*. Global is a synonym for non-personalized, Server is a synonym for centralized training. The experiment combinations are shown in Table 2.

**Model Training** Server-training uses SGD, while FL training uses Federated Averaging to combine SGD-trained client models (McMahan et al., 2016). Personalization in FL uses FURL training as described in section 3.5 The models were trained for 30 and 40 epochs for the Sticker and Subreddit datasets, respectively. One *epoch* in FL means the all samples in the training set were used once.

We ran hyperparameter sweeps to find the best model architectures (such as user embedding dimension, BLSTM and MLP dimensions) and learning rates. The FL configurations randomly select 10 users/round and run 1 epoch locally for each user in each round. Separate hyperparameter sweeps for FL and centralized training resulted in the same optimal embedding dimension for both configurations. The optimal dimension was 4 for the Sticker task and 32 for the Subreddit task.

| Config | With personalization | With FL |
|---|---|---|
| Global Server | No | No |
| Personalized Server | Yes | No |
| Global FL | No | Yes |
| Personalized FL | Yes | Yes |

Table 2: Experimental configurations.

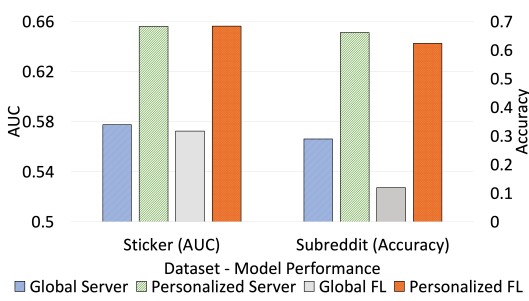

Figure 2: Performance of the configurations.

**Metrics**   We report accuracy for experiments on the Subreddit dataset. However, we report AUC instead of accuracy for the Sticker dataset since classes are highly unbalanced.

## 4.2   EVALUATION RESULTS

**Personalization improves the performance significantly.**   User embeddings increase the AUC on the Sticker dataset by 7.85% and 8.39% in the Server and FL configurations, respectively. The improvement is even larger in the Subreddit dataset with 37.2% and 50.51% increase in the accuracy for the Server and FL settings, respectively. As shown in Figure 2, these results demonstrate that the user representations effectively learn the features of the users from their data.

**Personalization in FL provides similar performance as server training.**   There is *no* AUC reduction on the Sticker dataset while the accuracy drops only 3.72% on the Subreddit dataset (as shown in Figure 2). Furthermore, the small decrease of FL compared to centralized training is expected and consistent with other results (McMahan et al., 2016). The learning curves on the evaluation set on Figure 3 show the performance of FL models asymptotically approaches the server counterpart. Therefore, FL provide similar performance with the centralized setting while protecting the user privacy.

**User embeddings learned in FL have a similar structure to those learned in server training.** Recall that for both datasets, the optimal embedding dimension was the same for both centralized and FL training. We visualize the user representations learned in both the centralized and FL settings using t-SNE (van der Maaten & Hinton, Nov 2008). The results demonstrate that similar users are clustered together in both settings.

Visualization of user embeddings learned in the Sticker dataset in Figure 4 shows that users having similar (e.g., low or high) click-through rate (CTR) on the suggested stickers are clustered together. For the Subreddit dataset, we highlight users who comment a lot on a particular subreddit, for the top 5 subreddits (*AskReddit*, *CFB*, *The_Donald*, *nba*, and *politics*). Figure 5 indicates that users who submit their comments to the same subreddits are clustered together, in both settings. Hence, learned user embeddings reflect users' subreddit commenting behavior, in both FL and Server training.

## 5   CONCLUSION AND FUTURE WORK

This paper proposes FURL, a simple, scalable, bandwidth-efficient technique for model personalization in FL. FURL improves performance over non-personalized models and achieves similar performance to centralized personalized model while preserving user privacy. Moreover, representations learned in both server training and FL show similar structures. In future, we would like to evaluate FURL on other datasets and models, learn user embeddings jointly across multiple tasks, address the cold start problem and personalize for users not participating in global FL aggregation.

| Config | AUC (Sticker) | Accuracy (Subreddit) |
|---|---|---|
| Global Server | 57.75% | 28.93% |
| Personalized Server | 65.60% | 66.13% |
| Global FL | 57.24% | 11.90% |
| Personalized FL | 65.63% | 62.41% |

Table 3: Performance of the configurations.

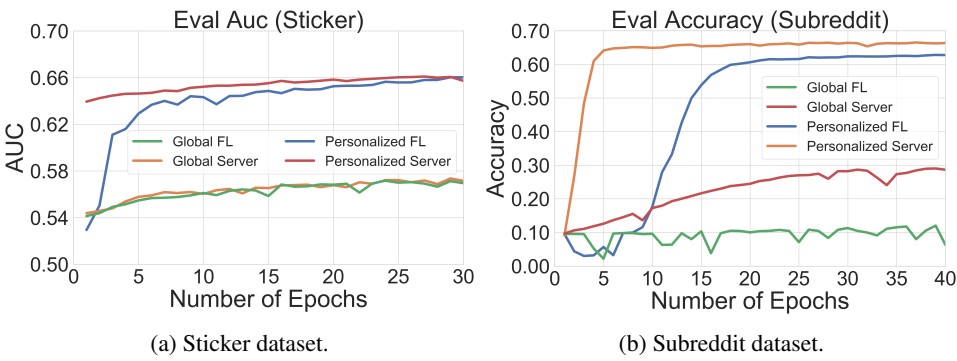

(a) Sticker dataset.

(b) Subreddit dataset.

Figure 3: Learning curves on the datasets.

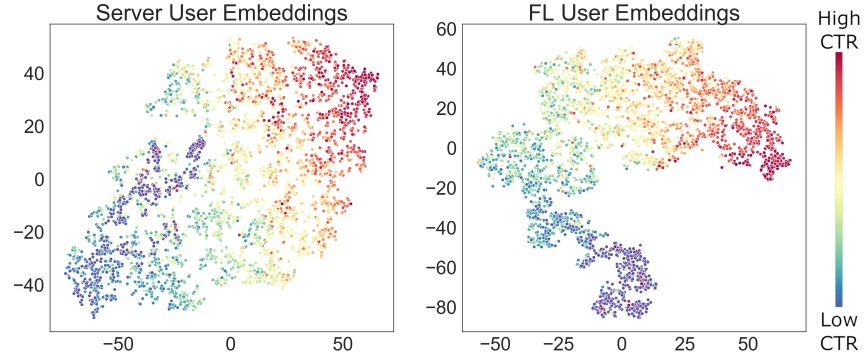

Figure 4: User embeddings in Sticker dataset, colored by user click-through rates (CTR).

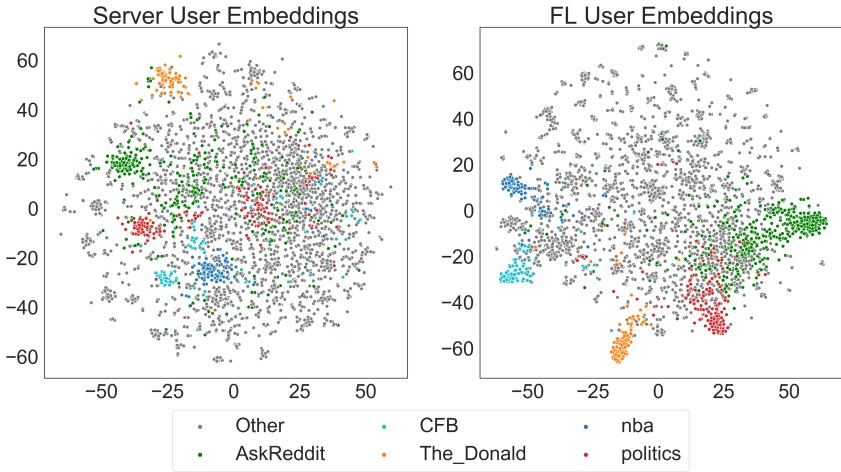

Figure 5: User embeddings in Subreddit dataset, colored by each user's most-posted subreddit.

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
