# OpenReview forum: "Federated User Representation Learning"
_ICLR.cc/2020/Conference — Reject_

### Official Review · AnonReviewer3 · 2019-10-20
**Official Blind Review #3**

**Rating:** 8

**Review:**

Authors proposed a formal training scheme (FURL) for personalized federated models. The claimed benefits of such models are 1) preservation of user privacy by keeping the personalized parameters locally on each user's device, and 2) reduced data exchange to make the training complexity grow linearly with the number of users. Authors approached the problem with defining the constraint of split personalization, and argued that common FL setting such as Federated Averaging could satisfy this constraint.

Authors designed a personalized classification deep network for two data sets, namely Stickers and SubReddit. Both tasks could benefit personal preference in addition to textual features: Stickers CTR depends on user's adoption of the feature; SubReddit categorization depends on user's past activities in each sub-Reddit. A clearly conducted experiment showed that personalization has a significant contribution in non-federated setting, and using FURL in federated setting achieved similar performance while non-personalized FL may suffer bigger loss (in the case of SubReddit). Authors also compared the conversage curve and visualized final embeddings to show that federated learning produces acceptable convergence and equally reasonable embeddings.

The paper is well written and all claimed contributions are well articulated. Reviewer didn't find any significant problems.

Reviewer has limited knowledge of previous work in the personalized FL field, thus is only able to confirm the novelty from Authors' related work section.

One comment about formatting: in Figure 5, the color dots in the legend could be larger for easier identification. Please also consider some color-independent label/description to help readers with difficulties in color perception. For example, you can name the color in the legend (i.e. "Red") and provide some text labels in the embedding chart to tell which part is mostly red.


**Experience Assessment:**

I do not know much about this area.

**Review Assessment: Checking Correctness Of Derivations And Theory:**

I carefully checked the derivations and theory.

**Review Assessment: Checking Correctness Of Experiments:**

I carefully checked the experiments.

**Review Assessment: Thoroughness In Paper Reading:**

I read the paper thoroughly.

---

> ### Comment · AnonReviewer3 · 2019-11-11
> **Additional notes after reading other anonymous reviews.**
>
> I did see the same concern raised by anon-reviewer #2 and #4 when reading the paper, but was not confident judging actual novelty since I'm less familiar with most recent FL research. So I chose to base on the author's own claim on their contrast with previous work.
>
> I'm very ok with adjusting it to weak accept and would lean more on the opinion of #2/#4 given their record of publication in the field.

---

### Official Review · AnonReviewer4 · 2019-10-23
**Official Blind Review #4**

**Rating:** 3

**Review:**

In this paper, the authors propose using federated learning (FL) to train personalized models, which improves the scalability and privacy preservation of the existing personalization techniques. The empirical results show good performance.

However, in general, I think the contribution is limited. The reasons are as follows:

1. The proposed algorithm, FURL, is a direct and simple combination of personalized model and FL. Although the authors claim that there is significant improvement in the performance, such improvement comes from the personalization. And, the personalization itself is not a novel thing (I think the personalized model used in this paper is similar to [1] or some other references. Please correct me if the personalized model used in this paper is new, since I'm not an expert in personalization.) Thus, in general, this paper simply use FL to replace fully synchronous SGD in the training of the personalized models. All the benefits claimed in the introduction, including scalability, privacy preservation, and improvement of performance, come from either vanilla personalization or vanilla FL. I fail to find any new contribution in this combination.

2. The authors emphasize a lot on the "independent aggregation constraint". Although it sounds like such constraint is designed especially for FL + personalization, it is actually a feature only for personalization, which has nothing to do with FL. Note that when doing inference/prediction, each user uses his/her own private part of the model. Different users' private part of models will never affect each other. It is equivalent to training a global model, which concatenates the private parts of models into a big model, and each user update the global model in a sparse manner. Thus, we can also train such personalized model with fully synchronous SGD with sparse gradients, which also does not synchronize the private parts. The private part is never shared by different users, no matter trained by fully synchronous SGD or FL.


------------
References

[1] Jaech, Aaron, and Mari Ostendorf. "Personalized language model for query auto-completion." arXiv preprint arXiv:1804.09661 (2018).

**Experience Assessment:**

I have published one or two papers in this area.

**Review Assessment: Checking Correctness Of Derivations And Theory:**

N/A

**Review Assessment: Checking Correctness Of Experiments:**

I assessed the sensibility of the experiments.

**Review Assessment: Thoroughness In Paper Reading:**

I read the paper thoroughly.

---

### Official Review · AnonReviewer2 · 2019-10-24
**Official Blind Review #2**

**Rating:** 1

**Review:**

This paper proposes the use of Federated Averaging for achieving personalised user embedding. Federated Learning is used whether they propose a particular split of model parameters with user embedding (private) and the overall BLSTM model (shared). Federated Averaging is used for the global update.

The key contribution of this paper is not clear. It seems to be the introduction of the notion of split-personalisation-constraint, and it shows that the modeling each user with a “private” embedding that feeds to a global MLP with a global BLSTM as another input (named as FURL) can achieve the constraint so that FL can be used. The originality is limited.


**Experience Assessment:**

I have published one or two papers in this area.

**Review Assessment: Checking Correctness Of Derivations And Theory:**

I assessed the sensibility of the derivations and theory.

**Review Assessment: Checking Correctness Of Experiments:**

I assessed the sensibility of the experiments.

**Review Assessment: Thoroughness In Paper Reading:**

I read the paper at least twice and used my best judgement in assessing the paper.

---

### Public Comment · ~Stone_Jamess1 · 2019-10-05
**Several questions about privacy concern and validity**

I have following questions

Privacy:
In this paper, the authors claim that local user will not update local embedding  but need to locally update the parameters. The update of the local parameters is jointly with the global shared parameters.
However, during the update process, the update of the global shared parameters is already influenced by the local private parameters and will carry somewhat information of the local private parameters. As a result, you cannot claim you protect privacy just because you didn't upload it.

Validity:
In the abstract, you write as: we show theoretically that this parameter split does not affect training for most model personalization approaches. Unfortunately, I didn't see any proof about it. Second, for federated setting, you have to update the parameters many steps before uploading. Even though you share all of the parameters, you will lose high accuracy. Now, you only share part of the parameters, you will lose much higher accuracy. So I think the result maybe "too good"

One more thing about writing:
I think the logic flow of this paper is rather mixed, and I spend lots of time to understand what the author is talking about even though I work on federated learning.

Thanks

---

> ### Comment · AnonReviewer4 · 2019-10-22
> **Interesting discussion**
>
> For privacy, I think the shared part follows the standard FL setting. The private part is never synchronized. So, in overall, the guarantee of privacy preservation is the same as the standard FL.
>
> For validity, I think the improvement of the accuracy comes from the personalization. As the experiments shows, personalization + fully synchronous training (I think it means the shared part is trained by fully synchronous SGD, while the private parts are still never synchronized) has better performance compared to personalization + FL.
> So, as you mentioned, FL does lose some accuracy.
> And, I guess you misunderstand "personalization". Basically, for personalization, each user has his/her own embedding layer, which is not shared by each other. And, this "personalization" is sth. orthogonal to FL (see the personalized language model: https://arxiv.org/pdf/1804.09661.pdf).
> Thus, the improvement comes from:
> 1. Since each user has his/her own user embedding layer, compared to the fully shared model, the personalized model has a much larger number of parameters. Thus, it is reasonable that the personalized model has better performance, due to the stronger representation power.
> 2. When testing on a new data, the model knows which user the data comes from, and use the corresponding user embeddings. The users' identity could be viewed as some extra information, which is not utilized by the fully shared model.
> I also believe that the personalized model has some limitations/tradeoffs. Basically, in testing, such personalization only works when the users already exist in the training data.

---

### Author Response · Authors · 2019-11-15
**Response to reviews**

Dear reviewers,

Thank you for the thorough reviews!

For the comments on novelty, although our method is not based on a complex theoretical ideas, it is simple yet practical. With minimal modification, the any personalization models which satisfy the split-personalization constraint can be used in FL in a scalable manner. We plan to change the title to "Embarrassingly Simple Approach to Personalization in Federated Learning" to emphasize this aspect.

In addition, we also plan to add a section to highlight the contrast with Federated Multi-Task Learning [1] and show that FURL is more scalable.

[1] Smith, Virginia, et al. "Federated multi-task learning." Advances in Neural Information Processing Systems. 2017.

---

### Decision · Program_Chairs · 2019-12-19

**Decision:**

Reject

**Comment:**

This manuscript personalization techniques to improve the scalability and privacy preservation of federated learning. Empirical results are provided which suggests improved performance.

The reviewers and AC agree that the problem studied is timely and interesting, as the tradeoffs between personalization and performance are a known concern in federated learning.  However, this manuscript also received quite divergent reviews, resulting from differences in opinion about the novelty and clarity of the conceptual and empirical results. Reviewers were also unconvinced by the provided empirical evaluation results.